# Identification and Functional Analysis of the *fruitless* Gene in a Hemimetabolous Insect, *Nilaparvata lugens*

**DOI:** 10.3390/insects15040262

**Published:** 2024-04-11

**Authors:** Biyun Wang, Zeping Mao, Youyuan Chen, Jinjun Ying, Haiqiang Wang, Zongtao Sun, Junmin Li, Chuanxi Zhang, Jichong Zhuo

**Affiliations:** State Key Laboratory for ManagingBiotic and Chemical Threats to the Quality and Safety of Agro-Products, Key Laboratory of Biotechnology in Plant Protection of MARA and Zhejiang Province, Institute of Plant Virology, Ningbo University, Ningbo 315211, China; biyun_wang@163.com (B.W.); 2111074030@nbu.edu.cn (Z.M.); 2211130005@nbu.edu.cn (Y.C.); 2011074054@nbu.edu.cn (J.Y.); 2011074045@nbu.edu.cn (H.W.); sunzongtao@nbu.edu.cn (Z.S.); lijunmin@nbu.edu.cn (J.L.); chxzhang@zju.edu.cn (C.Z.)

**Keywords:** *Nilaparvata lugen*, *fruitless*, *doublesex*, wing flapping, mating behavior

## Abstract

**Simple Summary:**

Mating behavior plays a crucial role in the survival and reproduction of insect populations. The *fruitless* (*fru*) gene, recognized for regulating male mating behaviors in *Drosophila melanogaster*, acts as a central “tuner”, shaping courtship behavior through sex-specific expression patterns within the courtship neural circuit. While *fru* homologs and sex-specific isoforms have been identified in other holometabolan insects, controlling mating behavior, hemimetabolous insects show the generation of non-sex-specific mRNAs by the *fru* gene, suggesting potential functional differences. This study focuses on *fru* homologs (*Nlfru*) in the Hemiptera species *Nilaparvata lugens*, utilizing RNAi-mediated knockdown to explore *Nlfru* functions in male mating behavior and tissue development. Our research contributes to understanding the regulation of mating behavior in *N. lugens* and sheds light on the evolution of the *fru* gene across insect species.

**Abstract:**

The *fruitless* (*fru*) gene functions as a crucial “tuner” in male insect courtship behavior through distinct expression patterns. In *Nilaparvata lugens*, our previous research showed *doublesex* (*dsx*) influencing male courtship songs, causing mating failures with virgin females. However, the impact of *fru* on *N. lugens* mating remains unexplored. In this study, the *fru* homolog (*Nlfru*) in *N. lugens* yielded four spliceosomes: *Nlfru-374-a/b*, *Nlfru-377*, and *Nlfru-433*, encoding proteins of 374aa, 377aa, and 433aa, respectively. Notably, only *Nlfru-374b* exhibited male bias, while the others were non-sex-specific. All NlFRU proteins featured the BTB conserved domain, with NlFRU-374 and NlFRU-377 possessing the ZnF domain with different sequences. RNAi-mediated *Nlfru* or its isoforms’ knockdown in nymph stages blocked wing-flapping behavior in mating males, while embryonic knockdown via maternal RNAi resulted in over 80% of males losing wing-flapping ability, and female receptivity was reduced. *Nlfru* expression was *Nldsx*-regulated, and yet courtship signals and mating success were unaffected. Remarkably, RNAi-mediated *Nlfru* knockdown up-regulated the expression of *flightin* in macropterous males, which regulated muscle stiffness and delayed force response, suggesting *Nlfru*’s involvement in muscle development regulation. Collectively, our results indicate that *Nlfru* functions in *N. lugens* exhibit a combination of conservation and species specificity, contributing insights into *fru* evolution, particularly in Hemiptera species.

## 1. Introduction

In insects, courtship behaviors are of paramount importance for the survival and reproduction of various insect populations, and these behaviors exhibit remarkable diversity. They include acoustic signals, licking, following, feeding, dancing, aggregation, and bioluminescence [1,2,3,4]. As a model organism, *Drosophila melanogaster* has been subject to deep investigation of its courtship behavior. During mating, the males vibrate one wing and produce a species-specific song for females [5,6]. The intricacies of this courtship display are predominantly regulated by specific regions within the central nervous system (CNS) and the associated expressed genes, which are regulated through sex-specific transcription factors that give rise to a sexually dimorphic CNS for sex-specific behaviors, providing the best understanding of the regulation between genes and the mating behavior [7].

In the male behavior of *D. melanogaster*, the gene *fruitless* (*fru*) is best studied, which has at least four promoters (P1–P4) and encodes 18 different isoforms [8,9,10,11,12]. The primary transcripts of promoter-1 are spliced in a sex-specific manner, producing female-specific and male-specific isoforms, *fruF* and *fruM*, which are controlled by Transformer (TRA) and Transformer2 (TRA2). Meanwhile, the other promoters generate non-sex-specific *fru-Com*, and all the isoforms of *fru* belong to the BTB-ZnF (BTB, a bric-a-brac domain; ZnF, zinc finger motif) family of transcription factors [9,10,11,12,13,14]. In males, the *fruM* isoforms encode proteins with C-terminal variants and have different functions in the mating behavior. The *fru-Com* isoforms, meanwhile, are expressed in both sexes and mediate the correct development of neuronal tissues. However, the functionalities of the noncoding female-specific *fruF* isoforms remain unknown [15]. Recent findings suggest that *fruM* specifies a sex circuitry that readily and specifically responds to conspecific females and generates robust courtship, which is not necessary for the generation of courtship behavior directed toward males. The temporal and spatial expression patterns of *fruM*, as well as its varying levels of expression, also contribute to the emergence of different courtship patterns observed in adult *D. melanogaster* [16,17]. 

Numerous studies have investigated the involvement of the *fru* gene in courtship behavior among various holometabolan insects. In the medfly (*Ceratitis capitata*), reducing the expression level of *fru* directly influences mating behavior, resulting in a prolonged courtship time and a decreased success rate of mating [18]. In the *Aedes aegypti* mosquito, *fru* mutant males fail to mate and gain strong attraction to a live human host [19]. In the silkworm (*Bombyx mori*), groups in which the *fru* gene is disrupted exhibit a significant increase in the time spent before mating compared to the control group, and male silkworms have slower and weaker mating behavior, causing a significant delay in locating female silkworms [20,21]. The conserved roles of *fru* in controlling courtship suggest that it acts as a master regulator of sexually dimorphic mating behaviors across holometabolan insects. Moreover, sex-specific splicing regulation of fru orthologs under TRA regulation has been found conserved in Diptera and Hymenoptera, but not in Orthoptera or Lepidoptera [22].

In *D. melanogaster*, *doublesex* (*dsx*) produces male- and female-specific isoforms, *dsxM* and *dsxF*. *dsxM* is involved in creating a sexually dimorphic CNS and is necessary for wild-type courtship in males. Both *dsxM* and *fruM* are found to be expressed in the same courtship-related neurons, and loss of *dsx* in males causes a reduction in the courtship level [23,24,25,26]. In *Nilaparvata lugens*, males signal to females using substrate-transmitted vibrations, and receptive females respond with a similar call of very variable duration [27]. Our prior investigations in *Nilaparvata lugens* revealed that the loss of *dsx* in this species led to male mating failures due to incomplete courtship songs compared to wild-type individuals [28]. This observation suggests that the *dsx* homolog in a Hemiptera species is also indispensable for the courtship song. However, whether *fru* homologs in Hemiptera species are also involved in the regulation of mating behavior, which includes the courtship song, wing flapping, and so on, has not been reported. In our current study, we identified the homologs of *fru* in *N. lugens* (*Nlfru*), the brown planthopper (BPH), and observed that *Nlfru* influences male wing-flapping ability during mating and the receptivity of females. Additionally, the expression of *Nlfru* is regulated by *Nldsx*. Furthermore, our findings indicate that *Nlfru* is implicated in the regulation of muscle development through the *flightin* gene. The interplay between *dsx*, *fru*, and associated genes underscores the conserved role of these factors in orchestrating complex mating behaviors across insect species.

## 2. Materials and Methods

### 2.1. Insect Husbandry

The BPH population used was collected in Hangzhou, China (30°16′ N, 12°11′ E) in 2008, and was raised in a growing chamber of 26 ± 1 °C, 80% relative humidity, and a 14 h light/10 h dark photoperiod.

### 2.2. Cloning of N. lugens Fruitless (Nlfru)

According to the manufacturer’s protocol, total RNA was isolated from BPHs using RNAiso Plus (TaKaRa, Dalian, China). The isolated total RNA (1 μg) was subjected to reverse transcription (RT) using Quant Reverse Transcriptase (TIANGEN, Beijing, China) in a 20 μL reaction, following the manufacturer’s instructions. The complete open reading frame (ORF) was obtained from the transcriptome sequences and validated using specific primers (see Appendix A). Subsequently, the PCR products were cloned into the PMD-19T vector (TaKaRa) and subjected to sequencing.

### 2.3. Semi-Quantitative RT-PCR

Total RNA was extracted from female (n = 10) and male (n = 15) BPH specimens. Following the manufacturer’s instructions, reverse transcription was performed using Quant Reverse Transcriptase (TIANGEN) in a 20 μL reaction volume. The cDNA was diluted 10-fold, and 1 μL was used as a template for subsequent PCR amplification. Gene-specific primers (see Appendix A) were used to amplify the specific splice forms of *fruitless*. The PCR conditions included an initial denaturation step at 94 °C for 5 min, followed by 35 cycles of denaturation at 95 °C for 30 s, annealing at 60 °C for 30 s, and extension at 72 °C for 10 min.

### 2.4. Real-Time Quantitative PCR (qPCR) Analysis

First, total RNA was extracted from BPHs using RNAiso Plus (TaKaRa). Then, the PrimeScript 1st Strand cDNA Synthesis Kit (catalog number 6110A, TaKaRa) was used to reverse transcribe each RNA sample (1 μg) into cDNA. The internal control for quantitative RT-PCR (qRT-PCR) was the Nl18S rRNA gene of the BPH (Nl18s-S: 5′-GTAACCCGCTGAACCT CCT-3′ and Nl18s-AS: 5′-tccgaagacctcactaatc-3′). The SYBR Premix Ex Taq Kit (TaKaRa) was used for qRT-PCR. The comparative threshold cycle method (ΔΔCT) was employed to assess quantitative changes [29], with wild-type samples used as negative controls.

### 2.5. RNA Interference (RNAi)

The target sequences of genes were cloned into the T-easy vector, with a length of around 500 bp. PCR was performed using the DNA template to generate a product that contained T7 promoter sequences at both ends, which was used for the synthesis of double-stranded RNA (dsRNA). dsRNA synthesis was carried out using a T7 High Yield Transcription Kit (Vazyme, city, China), following the manufacturer’s protocol. The dsRNA was quantified using a NanoDrop 2000 spectrophotometer (Thermo Fisher Scientific, Franklin Lakes, NJ, USA), where the concentration of dsRNA was about 2 μg/μL. The quality and size of the dsRNA were verified by 1% agarose gel electrophoresis. Next, we followed the efficient microinjection method described by Xue et al. [30]. to perform RNAi in brown planthoppers. We anaesthetized 3rd- or 5th-instar nymphs or newly emerged females (0–12th hour) of *N. lugens* by administering CO_2_ for 30 s and placed them on agarose plates. Then, about 10 nL or 100 nL dsRNA was microinjected into the body of *N. lugens*, and the RNAi efficiency was tested three days later.

### 2.6. Recording Courtship Behavior and Measurement of Courtship Signals

The mating behavior of BPHs was recorded using a video camera. Male and virgin female *N. lugens* individuals, both wild-type and those treated with ds*gfp* or ds*Nlfru*, were grouped separately. The insects were selected at the age of 3–4 days after eclosion. They were placed in cylindrical glass tubes containing rice seedlings, and their mating behavior was observed, including the wing-flapping ratio, the courtship duration, and the copulation percentage. More than 10 couples of each treatment were tested.

In order to detect the acoustic signals during the mating behavior of *N. lugens*, we utilized the vibration measurement and analysis system employed by Zhuo et al. [31]. Briefly, a pair of *N. lugens* was placed on a rice stem, and the mating signals were recorded using Adobe Audition (Adobe, San Jose, CA, USA). Subsequently, the data were analyzed using MATLAB (MathWorks, Natick, MA, USA) [32].

### 2.7. Lethality Statistics

Mortality statistics after nymph RNAi: At the 3rd instar, we administer ds*gfp* or ds*Nlfru* to the BPHs. After 24 h of treatment, we reared the BPHs in 3 groups with the same number in each. We counted and cleared the dead individuals every day until each remaining BPH had feathered.

Mortality statistics after maternal RNAi: To investigate the RNA interference in newly emerged females (0–1st hour), after they are allowed to mate with males on the third day, they were then raised individually. The females laid eggs for 6 days, after which hatched nymphs were collected and counted. Subsequently, the dead individuals were counted and cleared once a day until each nymph had emerged.

## 3. Results

### 3.1. The Homologs of Fruitless in N. lugens

To identify *fru* homologs in *N. lugens*, the *fru* of *D. melanogaster* (Gene ID: 42226) was used as a query to blast the data of our lab, including the genome, next-generation sequencing, and the third-generation full-length transcriptome of *N. lugens*. In total, four transcripts were identified, which encoded three kinds of proteins with 374aa, 377aa, and 433aa, named *Nlfru-374-a/b*, *Nlfru-377*, and *Nlfru-433* (Figure 1A). *Nlfru-374-a*, *Nlfru-377*, and *Nlfru-433* shared the exons 1, 3, and 4, and the semi-quantitative PCR with specific primers showed that they were non-sex-specific. Meanwhile, *Nlfru-374b* with specific exon 2 was found to be male-biased; however, *Nlfru-374a* and *Nlfru-374b* encoded proteins with the same sequences (Figure 1B and Appendix A).

The NlFRU proteins shared the same amino-terminal sequences with the conserved BTB domain, and the Znf domain was identified in both NlFRU-374 and NlFRU-377 (Figure 1C). The protein sequences of *fru* homologs in D. melanogaster, B. mori, and A. mellifera were also employed to align with three NlFRU proteins. The similarities of the conserved BTB domain between NlFRU and DmFRU, BmFRU, and AmFRU were 61.80%, 65.17%, and 68.54%, respectively; however, less conservation was found in the Znf domain and the rest sequences (Figure 1C). To elucidate the evolutionary relationship between the *fru* homolog identified in N. lugens and *fru* genes reported in other insects, we constructed a phylogenetic tree using the conserved BTB domains. Although the BTB domain of NlFRU was conserved with other insects, we found that when comparing those to D. melanogaster, *Nlfru* might have a greater evolutionary distance in *D. melanogaster* than in other species (Figure 2). All these results indicated that *Nlfru* might have different roles in *N. lugens*. 

### 3.2. Nlfru Controls the Wing-Flapping Behavior of Males and the Receptivity of Females

In the mating behaviors of *N. lugens*, female brown planthoppers will initiate abdominal tremors, and upon receiving the females’ signals, males will immediately run towards the females, engage in licking and wing-flapping behaviors, and complete mating within a few minutes (Figure 4). To study whether *Nlfru* affected the mating behavior of *N. lugens*, RNAi-mediated knockdown was employed in this study (Figure 1A). We performed RNA interference (RNAi) to knock down *Nlfru* with ds*Nlfru*, targeting common sequences in the third and fifth instars, and effectively reduced the expression of *Nlfru* within the *N. lugens* individuals (Figure 3A,B). Subsequently, we separately reared the treated males and wild-type virgin females and conducted mating experiments at 72–96 h after their eclosion. We found that with the loss of the *Nlfru* functions, some of the ds*Nlfru*-treated males lost the ability to flap their wings during mating with wild-type females (Figure 3D,E, Figure 4). 

To study which splicing isoform influenced the wing-flapping behavior, dsRNAs targeting the specific exons of *Nlfru* spliceosomes were injected into third-instar nymphs, and we found that the percentage of non-flapping individuals among the different dsRNA-treated males increased when compared with ds*gfp*-treated males; however, ds*Nlfru* targeting the common sequences caused more than 90% of males to lose their wing flapping in mating behavior (Figure 3D and Appendix A). Our results indicated that *Nlfru* might influence the wing flapping of males during mating with all the spliceosomes, and so ds*Nlfru* targeting the common sequences was used in the following research. However, we also found that non-flapping males could mate with females successfully, and there was no difference in the courtship duration between dsRNA-treated and control males (Figure 5A).

Since *Nlfru* is expressed in the early embryo stage, we used maternal RNAi to knock down the embryonic expression of *Nlfru*, and the RNAi effect lasted through all the development stages, including the embryo, nymph, and adult stages (Figure 3C). We found that over 80% of the male offspring did not flap their wings, a proportion higher than in the males treated with ds*Nlfru* in the third or fifth instar, and the earlier the instar chosen to knock down *Nlfru*, the higher the proportion of non-flapping males, suggesting that the function of *Nlfru* accumulated (Figure 3B). As *Nlfru* is expressed in both males and females, we also studied whether *Nlfru* influenced the females’ mating behavior. Because wing flapping is not the major feature of females’ mating behavior, the courtship duration and the copulated percentage were used to measure the influence of knocking down *Nlfru* with dsRNAs. ds*Nlfru* was administered to females during the third instar, after which they were mated with wild-type males. Our observations revealed that the courtship duration lasted approximately 10 min, showing no significant difference compared to the control group. Furthermore, over 80% of the BPHs successfully mated within 20 min (Figure 5A,C). However, the females of maternal RNAi offspring had a longer courtship duration in the mating with males, and only about half of the females could mate with males in 35 min, while the control females (wild-type or ds*gfp*-treated) could mate successfully in 20 min with wild-type, ds*gfp*-, or ds*Nlfru*-treated males. Finally, the injection of ds*Nlfru* in third-instar or newly emerged females (0–12th hour) did not influence the survival rate of BPH nymphs (Figure 5B,D and Appendix A). Our results indicated that *Nlfru* influences female receptivity, which might happen in the embryo stage.

### 3.3. Nlfru Does Not Affect the Courtship Songs

In the mating of BPHs, females and males usually meet at the same rice stem, and then they exchange mating signals. The mating signal of males consists of three parts, f: front vibrational frequency, consisting of irregular pulses, n: noncontiguous pulses, consisting of 1–5 discrete pulses, and m: main vibrational frequency, consisting of continuous regular pulses. Meanwhile, the female’s signal consists of continuous regular pulses (Figure 6A). In our previous study, we reported that *doublesex* in BPHs (*Nldsx*) controlled male courtship signals through its male-specific isoform, and ds*Nldsx*-treated males failed to mate with virgin females with abnormal acoustic signals [28]. In this study, we also found that *Nlfru* was regulated by *Nldsx*. In both sexes of ds*Nldsx*-treated BPHs, the expression of *Nlfru* was down-regulated, including the male-biased isoform (Figure 7 and Appendix A). However, we did not determine whether *Nlfru* was also involved in the regulation of acoustic signals of males.

In this study, we established that knocking down *Nlfru* in BPH males leads to the absence of wing flapping during courtship; however, the courtship duration is not influenced, indicating that females can receive the signals of non-flapping males. We collected acoustic signals from both wild-type male and female adults, as well as RNAi-treated male and female adults. The collected signals were analyzed based on three dimensions: waveform, pulse repetition rate, and dominant frequencies. Through our analyses, we determined the main spectrogram and the main power spectrum density of the waveforms of the acoustic signals for the treated group and the control group of females, indicating whether the males’ use or nonuse of wing flapping affected their acoustic signals (Figure 6 and Appendix A). Moreover, we removed the wings of male adults 3–4 days after eclosion, 12 h before mating with 3–4-day-old virgin females. We then recorded the courtship songs produced by the males, finding that there were no significant differences between the treated group and the wild-type control group in terms of the waveform, pulse repetition frequency, or dominant frequency of the acoustic signals (Figure 6). Therefore, we can conclude that wing vibration and acoustic signals are not directly related in brown planthoppers.

### 3.4. Nlfru Regulates the Expression of Flightin in Males

The gene *flightin* was initially identified in D. melanogaster and plays a major role in regulating muscle stiffness and the delayed force response during flight. In the brown planthopper, migratory BPHs of both sexes are long-winged morphs, whereas short-winged morphs are flightless, and *flightin* has been shown to interact with myosin and determine the sarcomere and muscle assembly length [33]. In the previous research, RNAi knockdown of *Nlfru* influenced wing flapping during the BPHs’ mating behavior, and we wondered whether *Nlfru* regulated the expression of *flightin* in male BPHs.

The *flightin* gene in the brown planthopper exhibits peak expression on the first day after eclosion. Therefore, we knocked down *Nlfru* in third-instar nymphs of the brown planthopper and collected and extracted RNA within 12 h after their eclosion. We then measured the expression level of *flightin*. Due to the dimorphic wing morphology in the brown planthopper, special attention was given during the sampling process to distinguishing between macropterous and brachypterous individuals. It is worth noting that *flightin* shows higher expression in macropterous brown planthoppers, which are long-winged morphs, while only a small level of expression is observed in male individuals of the short-winged brachypterous population (Figure 8). Interestingly, we found that in the ds*Nlfru*-treated macropterous males, the expression of *flightin* was up-regulated obviously, indicating that *Nlfru* might be involved in regulating muscle development.

## 4. Discussion

Mating behavior is pivotal for the survival and reproductive success of insect populations, with the *fru* gene being widely acknowledged for its role in regulating male mating behaviors across diverse insect species. In this study, we identified the sequences of *fru* homologs in *N. lugens*, a holometabolous species, and employed RNAi-mediated knockdown techniques to investigate the functional significance of *Nlfru*.

In this study, we found that of the four isoforms of *fru* identified in *N. lugens*, which are produced by alternative splicing, only one isoform, *Nlfru-374b*, is male-biased, not male-specific; moreover, *Nlfru-374a* and *Nlfru-374b* encode the same proteins, and all the transcripts encode proteins with the same BTB domain sequences. In grasshoppers *Chorthippus biguttulus*, *C. brunneus*, and *C. mollis*, the other three hemimetabolous species, the *fru* genes also generate non-sex-specific mRNAs [34]. In *Bemisia tabaci*, another hemimetabolous species, two *fru* transcripts start with the same BTB domain and are expressed in both sexes with different levels, though not in a sex-specific manner [35]. In the milkweed bug *Oncopeltus fasciatus*, only one transcript was identified, which affects the genitalia of both sexes [36]. However, in holometabolan insect species, *fruitless* regulates courtship behavior in various ways via its sex-specific isoforms, which are regulated by the sex determination pathway. In *D. melanogaster* and *C. capitata*, the *fru* pre-mRNA is regulated by alternative splicing and produces sex-specific isoforms, and the male-specific isoforms play critical roles in courtship regulation [10,18]. All these results indicate that *fru* homologs in hemimetabolous insects have a different expression strategy to those in holometabolous species.

In the mating behavior of *D. melanogaster*, males express a species-specific courtship song by extending and vibrating one wing, a behavior influenced by *fruM* transcripts [5,37,38]. The production of *fruM* transcripts is regulated differently in different tissues of *D. melanogaster*. In the central nervous system, the sex-specific alternative splicing of *fru* in females is regulated by *transformer* and *transformer2*, which are the sex-determination regulatory genes, and the male-specific isoform is produced by default. However, in the gonad stem cell niches, male-specific expression of *fru* is regulated by *dsx* and is independent of *tra* [39,40] In *N. lugens*, male wing vibration is similarly a distinctive feature during mating [41]. In this study, RNAi knockdown of *Nlfru* isoforms with different dsRNAs could influence wing flapping in the mating behavior to different degrees, where dsRNAs targeted the common sequences of all the isoforms that most influenced BPHs’ wing flapping, suggesting that all the isoforms of *Nlfru* are involved in the regulation of mating behavior. We also discovered that RNAi-mediated knockdown of *Nlfru* during the third-instar nymph or embryo stage, achieved through maternal RNAi, eliminates wing vibration behavior without affecting the courtship song of *N. lugens*. Subsequent experiments involved recording the courtship duration and success rate of wild-type males with removed wings, revealing that wing vibration behavior is not essential in the courtship repertoire of the brown planthopper. Our prior investigations demonstrated that RNAi-mediated knockdown of *Nldsx* influences the male courtship song, resulting in the disappearance of the main part (m1) and a failure to mate with females [28]. In this study, we further ascertained that *Nldsx* regulates the expression of *Nlfru* in both sexes, positioning *Nlfru* downstream in the sex determination pathway. Furthermore, we observed a reduction in female acceptance of courtship signals from males after maternal interference, as evidenced by an increase in courtship duration. However, interfering with females at the third-instar stage did not significantly affect their mating success rate or courtship duration, suggesting that *Nlfru* influences female acceptance during early developmental stages. Based on this, we propose a hypothesis that *Nlfru* regulates the neural differentiation in the central brain neurons of *N. lugens* females during the embryo stage, requiring further exploration in future studies.

In this study, we observed an up-regulation of the *flightin* gene’s expression by *Nlfru* in macropterous brown planthoppers; however, no discernible influence was detected in the brachypterous counterparts. Within the males of *N. lugens*, *flightin* expression occurs in the indirect flight muscle (IFM) of macropterous adults and the dorsal longitudinal muscle (DLM) in the two basal abdominal segments of both macropterous and brachypterous individuals [33]. NlFRU proteins, classified within the BTB-ZnF family, are predominantly characterized as transcriptional factors implicated in developmental functions. Notably, *fruM* directs the formation of the muscle of Lawrence, which is a male-specific muscle [9]. Accordingly, we speculate that *Nlfru* specifically regulates the expression of *flightin* in the IFM of macropterous individuals. Moreover, our findings indicate that *Nlfru* not only influences mating behavior and female receptivity through brain neurons but also plays a role in tissue development. 

In summary, our investigation employed RNAi-mediated knockdown to explore the *fru* homologs in *N. lugens*, a Hemiptera species. While *Nlfru* influences the wing-flapping behavior of males during mating, the males’ specific courtship song and the mating process itself remain unaffected. This implies that wing flapping is not a crucial element in the mating dynamics of *N. lugens*, distinguishing it from observations in *D. melanogaster*. Furthermore, the influences of *Nlfru* on female receptivity and tissue development show that the functions of *Nlfru* in *N. lugens* are extensive and complex, requiring more detailed research in the future.

## Figures and Tables

**Figure 1 insects-15-00262-f001:**
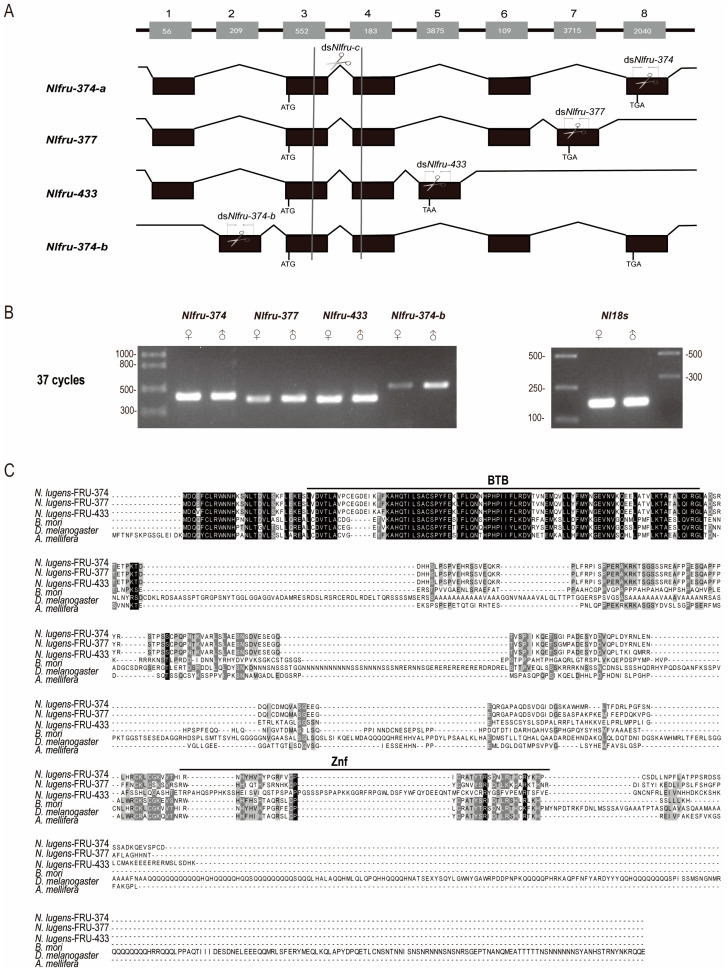
Sex-biased expression of *Nlfru* isoforms and their alternative splicing. (**A**) Boxes and lines denote exons and introns, respectively. The numbers in the boxes indicate the nucleotide numbers of exons. ATG sites and stop codons are indicated. Scissors indicate the RNAi regions on the isoforms. (**B**) Sex-biased expression of the four *Nlfru* splicing variants. In this RT-PCR, sex-specific cDNA was used as the template, and primer-374-a (*Nlfru-374-a*), primer-377 (*Nlfru-377*), primer-433 (*Nlfru-433*), and primer-374-b (*Nlfru-374-b*) were used to identify the respective four splicing variants; the primer *Nl18s*RNA was used as the positive control. (**C**) Sequence alignment of FRU proteins of *D. melanogaster*, *A. mellifera*, *B. mori*, and *N. lugens*. The black line indicates BTB and Znf, black letters indicate the conserved amino acid residues.

**Figure 2 insects-15-00262-f002:**
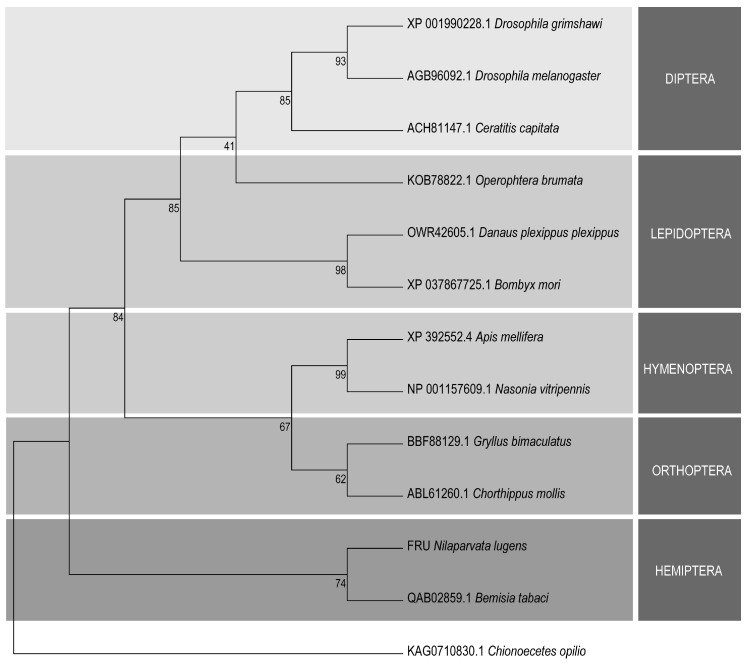
Phylogenetic analysis of the Fruitless proteins’ BTB sequence. The phylogenetic tree was constructed using MEGA v.5.05 maximum likelihood estimation. Bootstrap values are shown in the nodes. Branch lengths are proportional to sequence divergence.

**Figure 3 insects-15-00262-f003:**
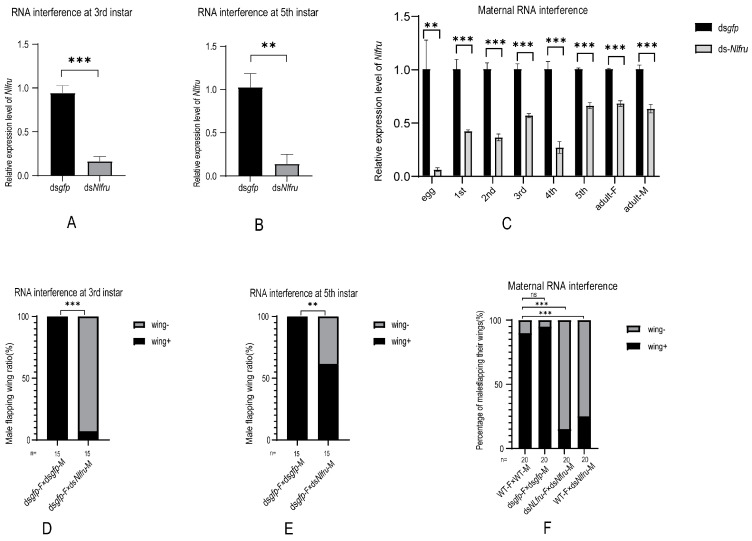
The expression of after RNAi and the proportion of flapping wings during the courtship of male BPHs. (**A**–**C**) *Nlfru* expression in ds*Nlfru*-treated BPHs. We performed 3 replications of each experiment, each containing 10 BPHs. (**A**) Silencing efficiency of fru after RNAi at 3rd instar. (**B**) Silencing efficiency of *Nlfru* after RNAi at 5th instar. (**C**) The expression of *Nlfru* in the offspring of the female parent after RNAi at various stages. (**D**–**F**) The male flapping-wing ratio of wild-type male brown planthoppers or those treated with *dsgfp* or ds*Nlfru* in different stages. The gray box represents the proportion of male BPHs that do not flap their wings during courtship (wing−), and the black box represents the proportion of their wings that flap during courtship (wing+). n is the number of experimental groups. The data results were generated using GraphPad Prism 5.0 for a chi-square test between the two groups. (**D**) Proportion of male flapping wings during courtship after RNAi at the 3rd instar. (**E**) Proportion of males flapping their wings during courtship after 5th-instar RNAi. (**F**) The proportion of male flapping wings during courtship after maternal RNAi. (Student’s *t*-test and the chi-square test were used, respectively: ** *p* < 0.01; *** *p* < 0.001; ns no significant).

**Figure 4 insects-15-00262-f004:**
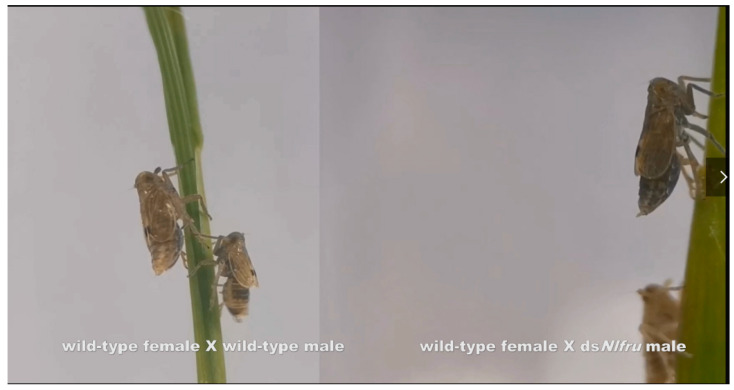
The mating behavior of wild-type BPHs and ds*Nlfru*-treated ones.

**Figure 5 insects-15-00262-f005:**
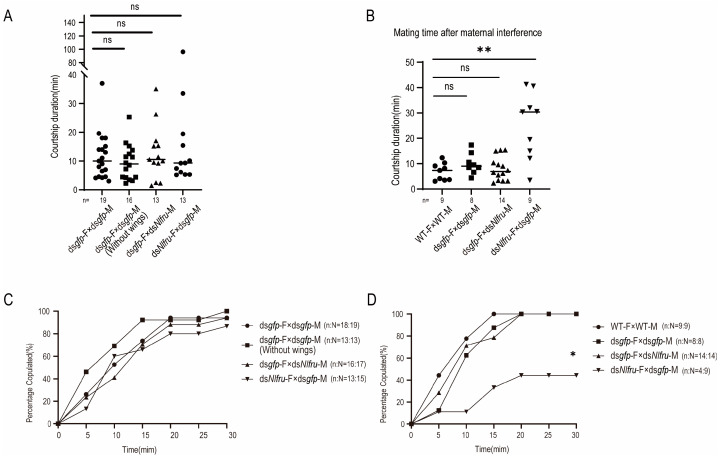
Courtship duration and courtship ratio. (**A**) After the 3rd-instar ds*Nlfru*, the time it takes for a female to accept a male courtship. (**B**) After the maternal RNAi, the courtship time of the offspring female. n represents the number of groups in (**A**–**C**) after the dsRNA knockdown in the 3rd instar, and the males’ success rate of mating at different periods. (**D**) After the maternal interference, the male offspring’s success rate of mating at different stages. ds*Nlfru* is targeted in the common sequences of all the transcripts. In Figure (**C**,**D**), N represents the total number of groups and n represents the number of mating groups. (Student’s *t*-test was used, ** *p* < 0.01; ns no significant. chi-squared test was used, * *p* < 0.01).

**Figure 6 insects-15-00262-f006:**
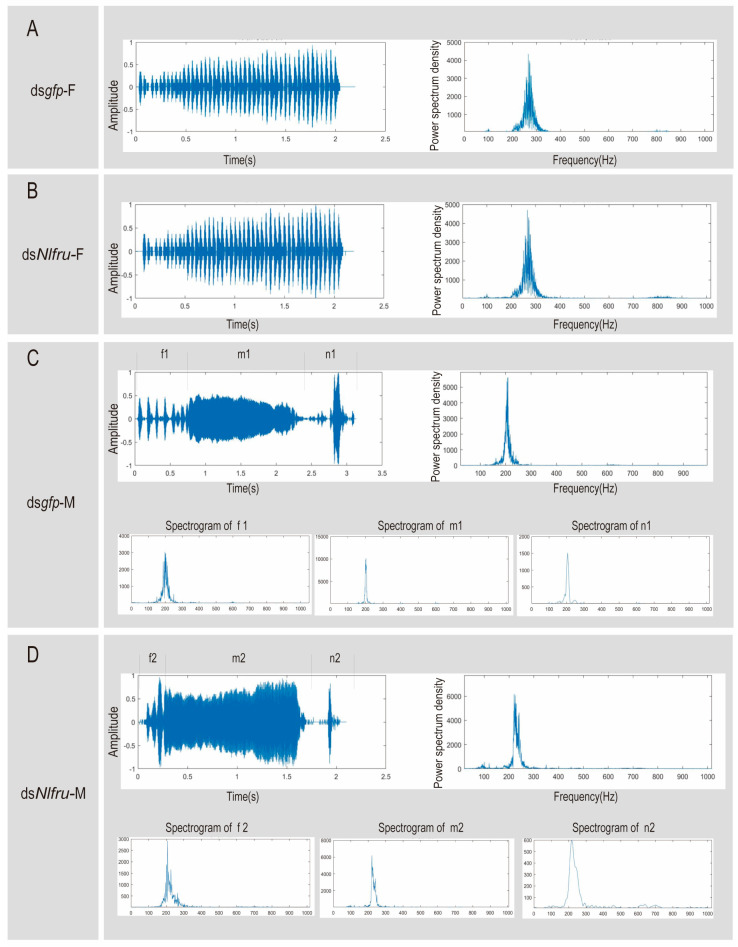
Waveforms and spectra of courtship signals in the BPH. (**A**) Waveforms and spectra of courtship signals in the wild-type females, where the main power spectrum density of the wild type is about 250 Hz. (**B**) Waveform and spectrum of the female courtship signal after ds*Nlfru*, where the main power spectrum density of the wild type is about 250 Hz. (**C**) Waveforms and spectra of courtship signals in the wild-type males, where the main power spectrum density of the wild-type is about 200 Hz; below are spectrograms for the f, m, and n segments, respectively, where the f, m, and n main spectrogram is about 200 Hz. (**D**) Waveform and spectrum of male courtship signals after ds*Nlfru*, where the power spectrum density of the wild type is about 200 Hz, and the f, m, and n main spectrogram is between 200 Hz and 300 Hz.

**Figure 7 insects-15-00262-f007:**
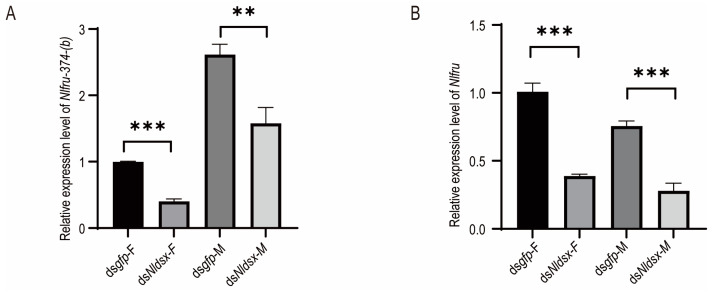
The expression of *Nlfru* after ds*Nldsx*. We performed 3 replications of each experiment, each containing 10 BPHs. (**A**) Expression of male-biased alternatively spliced *Nlfru*-374-b after ds*Nldsx*. (**B**) Expression of *Nlfru* after ds*Nldsx*. (Student’s *t*-test was used; ** *p* < 0.01; *** *p* < 0.001).

**Figure 8 insects-15-00262-f008:**
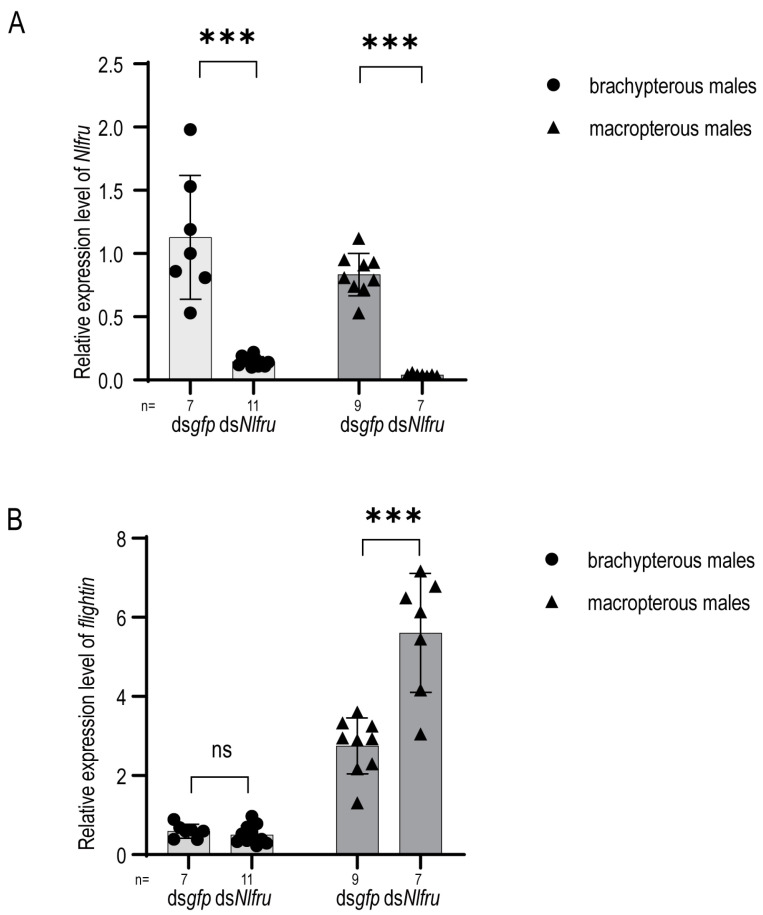
Silencing efficiency after ds*Nlfru* and the amount of *flightin* expression. (**A**) The expression of *Nlfru* in brachypterous males and macropterous males after RNAi was determined in BPHs, with the whole bodies of adult males used in this test. (**B**) The expression of *flightin* in brachypterous males and macropterous males after RNAi was determined in BPHs. n represents the number of groups in (**A**,**B**). (Student’s *t*-test was used; *** *p* < 0.001; ns no significant).

## Data Availability

Sequences of *Nlfru-374-a/b*, *Nlfru-377*, and *Nlfru-433* were deposited in GenBank with the accession numbers PP213269, PP213272, PP213270, and PP213271, respectively.

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
