# Peer review of "Identification and Functional Analysis of the fruitless Gene in a Hemimetabolous Insect, Nilaparvata lugens"

_insects, 2024, doi:10.3390/insects15040262_

Round 1

Reviewer 1 Report

Comments and Suggestions for Authors

The authors presented novel, interesting data concerning Nlfru structure, regulation, and function. Some relevant data descriptions are missing. The text has some mistakes and needs correction. 

Additional in silico analysis (head transcriptomes) are needed to evaluate if Nlfru has minor undetected splicing variants in this tissue. 

My suggestions:

1- Describe the Nilaparvata lugens sex determination pathway  XX/XY > NllFmd/NlFmd2-Nltra2>dsx.

As NllFmd RS_rich protein has a low short segment similarity to Apis CSD, and Apis CSD with Ceratitis TRA, NlFmd is likely functionally corresponding to CcTRA/DmTRA and  it is expected to interact with Nltra2 ti bind Nldsx pre-mRNA. Right?

2. Mention that sex-specific splicing regulation of fru orthologues under TRA regulation have been found conserved in Diptera, Hymenoptera,  but not in Orthoptera, Lepidoptera (see for example more recent reviews such as Saccone, G. 2022). Mention also the recent paper  “Distinct developmental mechanisms influence sexual dimorphisms in the milkweed bug Oncopeltus fasciatus”

https://www.ncbi.nlm.nih.gov/pmc/articles/PMC9890105/

3. Describe that the migratory BPHs of both sexes are long-winged morphs whereas short-winged morphs are obligate flightless. 

4 “ Notably, only Nlfru-374b exhibited male bias,…"

Please provide a measure of the quantity of the bias. 

5 In Fig. 1 B Nl-374-b is male-specific and not male-biased on gel analysis. Could you elaborate more info about the level of bias detected by qPCR (2 times difference) and not by electrophoresis? The two data are not coherent.

6 IN FIG. S1, panel A, the authors analysed wt-F and wt-M for Nlfru-374 isoform levels. Why the authors analysed the remaining 3 Nlfru isoforms in dsgfp-F and dsgfp-M individuals rather than wld type? Mislabelling?

7 Please correct legends (spliceosomes? -> Splicing isoforms)

Figure. S-1. The expression of different alternative spliceosomes in male and female insects. 354 

Figure. S-2. Silencing efficiency of specific RNAi for different alternative spliceosomes and the 355 proportion of males flapping their wings during courtship. A)-DExpression of Nlfru after 356 RNAi-specific selective spliceosome. E) Proportion of males flapping their wings during court-357 ship after RNAi on specific alternative spliceosomes

8 Clarify that you have asked if also fru splicing as dsx is under the control of NllFmd and that you have found rather that either fru transcription level or fru mRNA isoform stabilities are under the control of NldsxM and NldsxF in both sexes. 

Is Nl-374-b fru isoform male-bias under the control of NldsxM?

9 To identify fru homologs in N. lugens, the fru of D. melanogaster was used as a query to blast the genome and transcriptome of N. lugens: 

Please provide NCBI  links to access the genome and transcriptome data.

Is this one of the links?

https://www.ncbi.nlm.nih.gov/datasets/genome/?taxon=108931

10 As FruM mRNAs has low expression levels  because also expressed in the head, the authors could do a rather simple analyses, using SRA available data at NCBI to search for additional Nlfru isoforms in  transcriptomes of the head.

Here there are two SRA data from Head of Nl. I suggest that an assembly could reveal novel splicing isoforms of Nlfru expressed at low levels un the head, even sex-specific.

https://www.ncbi.nlm.nih.gov/sra/SRX10336750[accn]

https://www.ncbi.nlm.nih.gov/sra/SRX5008903[accn]

11 . In total, four transcripts were identified, 150 

which encoded three kinds of proteins with 374aa, 377aa, and 433aa, named Nlfru-374-a/b, 151 Nlfru-377, and Nlfru-433 (Fig.1A)

Please provide NCBI, Genbank, identification numbers for the seq data. 

12. “To study whether Nlfru affected the mating behavior of N. 179 lugens, RNAi-mediated knockdown was employed in this study (Fig.1A). We performed 180 RNA interference (RNAi) to knock down Nlfru with dsRNAs targeted common sequences 181 in the 3rd- and 5th- instar, and effectively reduce its expression within the N. lugens indi-182 viduals respectively (Fig.3A-a1, a2). Subsequently, we separately reared the treated males 183 and wild-type virgin females and conducted mating experiments during the 72-96 hours 184 after their eclosion. We found that with the loss of the Nlfru functions, some of the dsNlfru-185 treated males lost the ability to flap their wings during mating with wild-type females 186 (Fig.3B-b1, b2, Video1).” 

The authors should provide number of individuals used in the mating behaviour study, how many biological replicates have been used, the method for statistical analysis. All these data are missing in the material and methods as well as in the main text. I cannot find these data also in the legend of Figure 3.

13.” Proportion of male flapping wings during courtship after RNAi at the 3rd. b2) Proportion of males 196 flapping their wings during courtship after 5th instar RNAi. b3)The proportion of male flapping 197 wings during courtship after maternal RNAi. (**, P < 0.01; ***, P < 0.001). 198  “

What are the absolute numbers of males analysed? Data are missing. 

14. the percentage of non-flapped ones 

Please change into non-flapping ones.

15 non-flapped males 

Please change into  non-flapping males.

16 Considering that Nlfru expressed in early embryo 

Considering that Nlfru is expressed in early embryo 

16bis: Figure 4. Courtship duration and courtship ratio.

Please add in the legends and or in the math methods the numbers of individuals used and the statistical methods.

16 tris

Figure 5. The expression of Nlfru after dsNldsx. A) Expression of male-specific alternatively spliced Nlfru-374-b after dsNldsx. B) Expression of Nlfru after dsNldsx. (**, P < 0.01; ***, P < 238 0.001). 

It is seriously confusing that the authors wrote here in the legend “Expression of male-specific alternatively spliced Nlfru-374-b”, while in the main text they concluded that Nlfru Nlfru-374-b  is NOT sex-specifically splicing but only male-biased, athough the gene electrophoresis supports the sex-specific splicing and the qPCR the male-bias.

I invite the authors to be more focused and precise in analysing their data dn writing the article. They should correct the various point accordingly to their observations, data and conclusions. If data are contradictory the authors should report them and explain their doubts, hypothesis on potential technical problems.

Fig. 4 B) Expression of Nlfru after dsNldsx. The authors should specific which Nlfru isoform has been evaluated here.

17 “Considering that Nlfru expressed in early embryo stage, we used maternal RNAi to 217 knock down the embryonic expression of Nlfru, and the RNAi effect lasted all the devel-218 opment stage, including the embryo, the nymph, and the adult stages (Fig.3A-a3). We 219 found that over 80% of the offspring males did not flap their wings, the percentage of 220 which was higher than the ones treated with dsNlfru in 3rd or 5th instar, and the earlier 221 instar to knock down Nlfru, the higher proportion of non-flapped males, suggesting that 222 the function of Nlfru was accumulated (Fig.3B) “

In D. melanogaster loss of function of fru during development leads to lethality. The authors missed to show in a Table if there is any lethality difference when treating at embryonal or larval stages the individuals. Please describe the experiments with numbers and tables and the statistical methods.

17 bis

“The mating signal of males consists three parts: f1 con-241 sists of irregular pulses, n1 consists of 1-5 discrete pulses, and m1 consists of continuous 242 regular pulses, and the female’s signal also consists of continuous regular pulses . “

What are the pulses? How are they generated by males and females?

The authors should explain/described to the reader the wavefoms and spectra of courtship signals in normal insects. Some background will be useful for the reader.

How do the wingless males produce courtship songs?

18 “Figure 6. Wavefoms and spectra of courtship signals in the BPH. A) Wavefoms and spectra of court-251 ship signals in the wild-type females. B) Waveform and spectrum of female courtship signal after 252 dsNlfru. C) Wavefoms and spectra of courtship signals in the wild-type males; Below are the spec-253 trograms for the fm and n segments respectively. D)Waveform and spectrum of male courtship 254 signals after dsNlfru. “

The authors should provide description of biological replicates and number of analysed individuals to this experiment and the statistical methods.

19 Gene flightin, was initially identified in D. melanogaster and plays a major role in regulat-272 ing muscle stiffness and the delayed force response during flight. 

Please provide a reference here. Is flightin expression under the control of fru or dsx in Drosophila?

20 “Therefore, it is reasonable to speculate that whether Nlfru regu-275 lated wing oscillation behavior in male insects.” 

Please rewrite the incomplete sentence.

21 please rewrite the following not very clear sentence .

dsRNAs. The females injected with dsNlfru in 3rd instar mated with wild-type males,

 227 and we found that the courtship duration was about 10 minutes, which was no different 228 from the control ones, and more than 80% BPHs could mate successfully in 20 minutes 229 (Fig. 4A-a1, B-b1). 

22. “Fig. 7. Silencing efficiency after dsNlfru and the amount of flightin expression. AThe expression of 278 Nlfru in brachypterous males and macropterous males after RNAi was performed on BPHs. BThe 279 expression of flightin in brachypterous males and macropterous males after RNAi was performed 280 on BPHs. (**, P < 0.01; ***, P < 0.001). “ Please describe the statistical methods.

Please provide numbers of individuals used in the text/meth meds and precise description of the experiment data and statistics.

The Nlfru expression levels in brachypterous males and macropterous males are similar Fig. 7A. The expression of flightin in brachypterous males and macropterous males are different. If Nlfru controls the higher level of flightin expression in macropterous males, we could expect higher levels also of Nlfru in macropterous males. But this is not the case. How the authors could try to explain this apparent inconsistency? 

In the fourth column Fig. 7B, the 4 individuals used for the analyses show very strong differences in their values. Which statistical method has been used? I fear that the few individuals used do not conclusively support the conclusion of the authors concerning flightin increased expression in dsNlfru treated individuals. 

23 “Therefore, we knocked down Nlfru in third-instar nymphs of the brown 283 planthopper and collected and extracted RNA within 12 hours after their eclosion.”

How many individuals? How many replicates? Please describe it in the text or legends.

Discussion

 Discuss also your findings referring also to the following article:

https://pubmed.ncbi.nlm.nih.gov/33788836/

Doublesex regulates fruitless expression to promote sexual dimorphism of the gonad stem cell niche.

These authors found that  “Unlike previously studied regulation of sex-specific Fru expression, which is regulated by alternative splicing by Transformer (Tra), we show that male-specific expression of fru is regulated downstream of dsx, and is independent of Tra. Regulation of fru by dsx also occurs in the nervous system. fru genetically interacts with dsx to support maintenance of the hub throughout development” 

The authors should correct the following sentences.

- Gene Nlfru is the downstream genes of the sex-determination pathway, 347 Nldsx regulates the expression of Nlfru in both sexes, suggesting that Nldsx isoforms in 348 females also are functional. 

-… which needs more detail researches in future. 

Comments on the Quality of English Language

none

Reviewer 2 Report

Comments and Suggestions for Authors

See attached word file

Comments on the Quality of English Language

See general comments for details. But extensive editing of the language is needed as in many cases it hinders correct interpretation of the text.

Reviewer 3 Report

Comments and Suggestions for Authors

In the manuscript "Identification and functional analysis of the fruitless gene in a hemimetabolous insect, Nilaparvata lugens", Wang et al present a study in which they investigated the function of fru homologs (Nlfru) in the Hemiptera species Nilaparvata lugens. They performed RNAi-mediated knockdown of the Nlfru genes and found that embryonic knockdown causes males losing wing-flapping behavior during mating, and females reduced receptivity. They also found that Nlfru expression was regulated by Nldsx gene, and it then regulates flightin expression in some males. Overall, the study is well designed, and their conclusion is supported.

However, I have a few concerns listed below:

1.     The authors mentioned a few times in the Results and Discussion section about their previous finding about the Nldsx (For example, from line 244-246, 317-319) without proper citation. I suggest the authors to check carefully to make sure all the previous work mentioned in the manuscript have proper references.

2.     Some of the results do not have associated statistics, for example Figure 3B and Figure 4B. The authors should perform statistics test with these results and label if the changes are significant. For the statistics done in the manuscript, the authors should state what statistics test and/or correction was done to get the p-values in the figure legends.

3.     I suggest that authors bring Figure 6 before Figure 5. I n the section “3.3 Nlfru does not affect the courtship songs” authors talk about the influence of Nlfru on courtship songs, it makes more sense to present “Figure 6. Wavefoms and spectra of courtship signals in the BPH” first in this section. Then present Figure 5 and Figure 7 afterwards.
